# Using Unmanned Aerial Vehicle and Ground-Based RGB Indices to Assess Agronomic Performance of Wheat Landraces and Cultivars in a Mediterranean-Type Environment

Rubén Rufo [1], Jose Miguel Soriano [1], Dolors Villegas [1], Conxita Royo [1] and Joaquim Bellvert [2,*]

[1] Sustainable Field Crops Programme, IRTA (Institute for Food and Agricultural Research and Technology), 25198 Lleida, Spain; ruben.rufo@irta.cat (R.R.); josemiguel.soriano@irta.cat (J.M.S.); dolors.villegas@irta.cat (D.V.); conxita.royo@irta.cat (C.R.)
[2] Efficient Use of Water in Agriculture Programme, IRTA (Institute for Food and Agricultural Research and Technology), 25198 Lleida, Spain
* Correspondence: joaquim.bellvert@irta.cat; Tel.: +34-973032850 (ext. 1566)

**Abstract:** The adaptability and stability of new bread wheat cultivars that can be successfully grown in rainfed conditions are of paramount importance. Plant improvement can be boosted using effective high-throughput phenotyping tools in dry areas of the Mediterranean basin, where drought and heat stress are expected to increase yield instability. Remote sensing has been of growing interest in breeding programs since it is a cost-effective technology useful for assessing the canopy structure as well as the physiological traits of large genotype collections. The purpose of this study was to evaluate the use of a 4-band multispectral camera on-board an unmanned aerial vehicle (UAV) and ground-based RGB imagery to predict agronomic traits as well as quantify the best estimation of leaf area index (LAI) in rainfed conditions. A collection of 365 bread wheat genotypes, including 181 Mediterranean landraces and 184 modern cultivars, was evaluated during two consecutive growing seasons. Several vegetation indices (VI) derived from multispectral UAV and ground-based RGB images were calculated at different image acquisition dates of the crop cycle. The modified triangular vegetation index (MTVI2) proved to have a good accuracy to estimate LAI ($R^2$ = 0.61). Although the stepwise multiple regression analysis showed that grain yield and number of grains per square meter ($NGm^2$) were the agronomic traits most suitable to be predicted, the $R^2$ were low due to field trials were conducted under rainfed conditions. Moreover, the prediction of agronomic traits was slightly better with ground-based RGB VI rather than with UAV multispectral VIs. NDVI and GNDVI, from multispectral images, were present in most of the prediction equations. Repeated measurements confirmed that the ability of VIs to predict yield depends on the range of phenotypic data. The current study highlights the potential use of VI and RGB images as an efficient tool for high-throughput phenotyping under rainfed Mediterranean conditions.

**Keywords:** high-throughput phenotyping; drought stress; UAV imagery; ground-based RGB image; vegetation indices; phenology; grain yield; biomass

## 1. Introduction

Wheat is the main crop around the world and provides 18% of the global human intake of calories and 20% of protein (http://www.fao.org/faostat/ accessed on 14 December 2020). Since global wheat demand is predicted to increase by 60% by the year 2050, there is an urgent need to raise wheat production by 1.7% per year until then [1]. Therefore, the rate of genetic improvement required in the next decades is higher than that achieved so far [2]. Given the limitations imposed by the soil availability for agricultural uses, most increases rely on the release of improved cultivars with enhanced yield potential and stability under variable environmental conditions. Drought stress during the grain filling period, originating from a combination of water deficit and high temperatures, is the main

constraint on wheat yield in semi-arid environments, such as the Mediterranean Basin [3], which has been identified as one of the regions most sensitive to the effects of climate change. A reduction of 20% in yearly precipitation and a mean temperature increase of 4 °C have been predicted for this area by climate change models (http://www.ipcc.ch/ accessed on 14 December 2020) [4]. For this reason, breeding programs are focusing on the adaptability and stability of new cultivars that can be successfully grown in dry areas [5].

There is a general agreement that phenotyping is currently the bottleneck for further yield increases in breeding programs [6]. The availability of cost-effective technologies able to phenotype large number of plots in a rapid, cost-effective, and high spatial resolution way is essential for genetic progress [7]. In recent years, high-throughput phenotyping (HTP) has been increasingly used in plant breeding to estimate traits such as yield, green biomass, plant height, and leaf area index (LAI) [8–10]. Among the different approaches used for field HTP, remote sensing permits nonintrusive, nondestructive, high-throughput monitoring of agronomic, physiological, and architectural plant traits [11]. In HTP, this approach is mostly through spectral vegetation indices (VI), which are obtained from the formulation of different wavelengths mostly located at the visible, red-edge, and near-infrared [12]. Usually, these indices are calculated from multispectral cameras installed on-board an unmanned aerial vehicle (UAV), with the main advantage being the capacity for screening hundreds of plots in a short period of time [13,14]. Various authors have stressed the suitability of using VI measured early in the season for grain yield forecasting [15], although anthesis and milk grain development have been shown to be more useful for yield appraisal in wheat [16,17]. Some of them have shown a root mean square error (RMSE) ranging from 0.57 to 0.97 t/ha for predicting yield in wheat [18,19]. Other methodologies also use machine-learning regressions, chemometrics, radiative transfer models, photogrammetry, or hybrid approaches to estimate vegetation traits [20–22]. On the other hand, far-infrared (thermal) radiation and LIDAR sensors have been respectively used to estimate plant water status [23] and to characterize the architectural features [24].

Red-green-blue (RGB) imagery, obtained from conventional digital cameras, has also been reported to be a suitable method to calculate vegetation indices for wheat breeding in water-limited environments [25]. Conventional digital cameras are more affordable, portable, and easy to use, being a cost-effective way to obtain images of a large number of samples with minimum effort [26]. Moreover, their use has also been proposed in breeding programs for assessing plant traits such as green biomass since the calculation of vegetation indices is based on simple methods that can obtain data automatically from a high number of images [25]. Some studies have demonstrated that vegetation indices derived from RGB cameras are also able to give the same or better results as those obtained from multispectral images [9,27]. Kefauver et al. [27] compared UAV and field-based high-throughput phenotyping using RGB cameras for assessing nitrogen use efficiency (NUE) in barley. It was found that the regression models explained 77.8% and 71.6% of the variance in yield from UAV and ground data, respectively, while combining the datasets led to an increase in the explanation of variance to 82.7%. Gracia-Romero et al. [9] compared the performance of RGB images acquired from ground and aerial cameras to estimate yield in maize under different levels of phosphorus fertilization. The authors found that, in general, ground-based RGB indices correlated in a comparable way with grain yield.

Most studies comparing the performance of RGB and multispectral images for the assessment of wheat traits have been conducted on sets of semidwarf cultivars grown in well-irrigated fields, where the expression of the yield potential and the range of phenotypic values are maximized, or under different irrigation treatments [28]. However, information is lacking regarding the suitability of remote sensing images to predict agronomic traits of wheats with contrasting canopy architectures under rainfed conditions. The current study examines the performance of VIs obtained at different dates from a 4-band multispectral camera (Parrot Sequoia) on-board UAV and those obtained from ground-based RGB images to assess agronomic traits of large panels of bread wheat landraces and modern cultivars adapted to Mediterranean conditions.

## 2. Materials and Methods

### 2.1. Experimental Field Setup and Agronomic Data Recording

A collection of 365 bread wheat (*Triticum aestivum* L.) genotypes from the MED6WHEAT IRTA-panel [29] was used in this study. The collection consisted of 181 landraces and 184 modern cultivars from 24 and 19 Mediterranean countries, respectively (Table S1). Field experiments were conducted at Gimenells, Lleida (41°38′ N and 0°22′ E, 260 m a.s.l) under rainfed conditions for two consecutive growing seasons, 2016–2017 and 2017–2018. Experiments followed a nonreplicated augmented design with two replicated checks (*cv.* 'Anza' and 'Soissons') and plots of 3.6 m$^2$ (1.2 m wide × 3 m long) with eight rows spaced 0.15 m apart. The seed rate was adjusted to 250 germinable seeds per m$^2$ and the plots were kept free of weeds and diseases. The sowing dates were 21 November 2016 and 15 November 2017.

Phenology was assessed based on the scale of Zadoks et al. [30]. A growth stage (GS) was considered to have been achieved when at least 50% of the plants reached it. The following six GS were determined at each plot: stem elongation or when the first node was detectable (S, GS31); booting, determined when boots swollen (B, GS45); heading (H, GS55); anthesis (A, GS65); medium milk-grain development (M, GS75); and hard-dough grain development (D, GS87). Meteorological data were recorded from a weather station placed in the experimental field.

The following agronomic traits were measured: yield, biomass, number of spikes per square meter (NSm$^2$), number of grains per square meter (NGm$^2$), and thousand kernel weight (TKW). The NSm$^2$, NGm$^2$, and TKW were obtained from samples collected at maturity one week before harvest from 1-m-long central row of each plot. After harvesting, plants were stored in a glasshouse in paper sacks at room temperature during five months until processing. Subsequently, samples were processed as dry matter after drying them at 70 °C for 24 h to determine the aboveground biomass (t/ha). The plots were mechanically harvested at maturity, and the grain yield (GY, t/ha) is expressed on a 12% moisture basis.

The fraction of intercepted photosynthetically active radiation (fiPAR) was measured from 13:00 to 15:00 (local time) at each image acquisition date in 64 different plots of each landrace and modern set of genotypes using a portable ceptometer (AccuPAR model LP-80, decagon devices Inc., Pullman, WA, USA). Measurements were collected in clear sky conditions. Two measurements per plot were recorded by placing the ceptometer in a horizontal position at ground level. A fixed tripod connected to the sensor allowed us to collect the incident radiation above the plants. These measurements were also used to obtain the leaf area index (LAI) using the Norman-Jarvis model [31], and assuming a leaf area distribution parameter for wheat as 0.96.

### 2.2. Remote Sensing Images Acquisition

During the first growing season, both ground-based RGB and multispectral UAV images were acquired on the following three dates: 28 March (128 days after sowing, DAS); 21 April (151 DAS), and 19 May (179 DAS). Figure 1 shows the color of the different genotypes in the field at the three image acquisition dates. The adverse meteorological conditions during the spring of the second year hindered image capturing at the early growth stages. Therefore, images were collected on April 17 (153 DAS) and May 18 (184 DAS), to match the main growth stages of the crop. Table 1 summarizes the growth stages of the genotypes included in the panel at each image acquisition occasion.

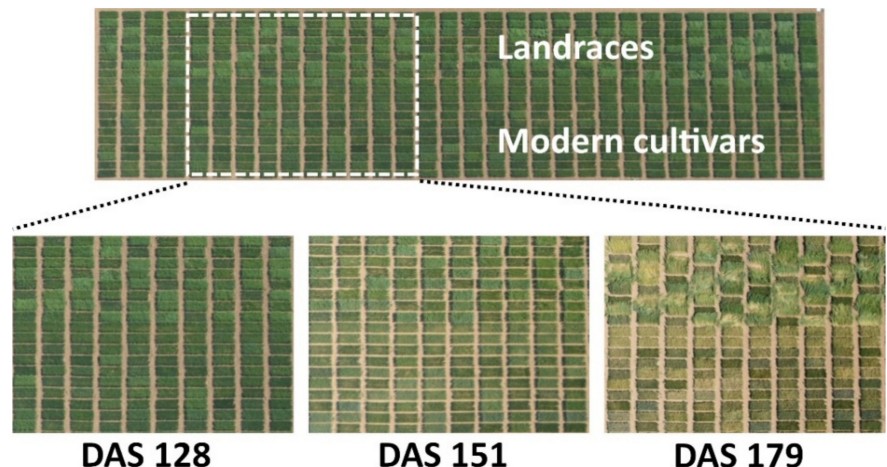

**Figure 1.** Field view of both collection sets, landraces and modern cultivars, at each image acquisition date of the growing season 2016–2017. DAS, days after sowing.

**Table 1.** Number and percentage of genotypes showing each growth stage at each image acquisition occasion.

| Date | Days after Sowing | Growth Stage | Number of Genotypes | (%) |
|---|---|---|---|---|
| **Landraces 2016–2017** | | | | |
| 28 March 2017 | 128 | Stem elongation | 181 | 100 |
| 21 April 2017 | 151 | Booting | 95 | 53 |
| | | Heading | 53 | 29 |
| | | Anthesis | 29 | 16 |
| | | Milk development | 4 | 2 |
| 19 May 2017 | 179 | Milk development | 52 | 29 |
| | | Dough development | 129 | 71 |
| **Modern 2016–2017** | | | | |
| 28 March 2017 | 128 | Stem elongation | 169 | 92 |
| | | Booting | 15 | 8 |
| 21 April 2017 | 151 | Booting | 8 | 4 |
| | | Heading | 72 | 39 |
| | | Anthesis | 45 | 25 |
| | | Milk development | 59 | 32 |
| 19 May 2017 | 179 | Dough development | 184 | 100 |
| **Landraces 2017–2018** | | | | |
| 17 April 2018 | 153 | Stem elongation | 97 | 53 |
| | | Booting | 83 | 46 |
| | | Heading | 1 | 1 |
| 18 May 2018 | 184 | Milk development | 109 | 60 |
| | | Dough development | 72 | 40 |
| **Modern 2017–2018** | | | | |
| 17 April 2018 | 153 | Stem elongation | 26 | 14 |
| | | Booting | 126 | 69 |
| | | Heading | 32 | 17 |
| 18 May 2018 | 184 | Milk development | 66 | 36 |
| | | Dough development | 118 | 64 |

### 2.2.1. Ground-Based RGB Vegetation Indices

Ground-based RGB images were collected in clear-sky conditions from 12:00 to 14:00 (local time) over the two years at the same day as UAV multispectral image acquisition. Ground-based RGB images were taken following the methodology reported by Casadesús

and Villegas [26]. A digital camera (Sony Alpha A5000, TYO, JPN) was used, with an objective Sony 16–50 mm at the minimum focal length, 19.8 megapixels of resolution, fixed aperture of F3.5, shutter speed of 1/250, without flash, and the aperture in automatic. When the plants were shorter than 120 cm, pictures were taken by holding the camera at 150 cm, approximately 50 cm from the border of the plot and oriented downwards. Once the average plot height exceeded 120 cm (which was the case with some landraces), it was necessary to use a camera stick at 170–190 cm. Three pictures were obtained per plot without stopping, covering the central rows of each plot in a zenithal plane. All the images were 1152 × 768 pixels, saved in JPEG format and processed with open-source BreedPix v0.2 software [25]. RGB indices were calculated based on properties of color related to the "greenness" of the canopy. Ten vegetation indices (VIs) were calculated following the protocol described in Casadesús et al. [25] (Table 2). As described in Kefauver et al. [27], hue, intensity, and saturation are the components of the HIS (hue–intensity–saturation) color space. Similar to intensity is the parameter lightness in both CIE-Lab and CIE-Luv color spaces, defined by the Commission Internationale de l'Éclairage (CIE), where a* and u* represent a color in an axis from green to red and b* and v* from yellow to blue according to the human visual system.

**Table 2.** Red-green-blue (RGB) vegetation indices, based on different color properties, used in the study.

| Parameter | Definition | Reference |
|-----------|------------|-----------|
| Intensity | Brightness of the image from black to white | |
| Hue | Color tint | |
| Saturation | Amount of tint | |
| Lightness | Overall albedo from the HIS color space | |
| a* | | [32] |
| u* | Red-green spectrum of chromaticity | |
| b* | | |
| v* | Yellow-blue color spectrum | |
| GA | Green area | [25] |
| GGA | Greener area | |

a* and u* represent a color in an axis from green to red and b* and v* from yellow to blue according to the human visual system.

### 2.2.2. Multispectral Images Acquired with the UAV

The UAV used for the multispectral image acquisition was the DJI S800 EVO hexacopter (Nanshan, CHN) (Figure 2a). Flight altitude was 40 m above ground level (AGL). The multispectral camera used was a Parrot Sequoia (Parrot, Paris, France) with a 1.2 mega-pixel sensor yielding a resolution of 1280 × 960 pixels. Horizontal, vertical, and diagonal field of view (HFOV, VFOV, and DFOV) provided by the optical focal length were 61.9°, 48.5°, and 73.7°, respectively. The camera included four individual image sensors with filters centered at the wavelengths and full-width half-max bandwidths (FWHM) of 550 ± 40 (green), 660 ± 40 (red), 735 ± 10 (red edge) and 790 ± 40 nm (near infrared), respectively. The Parrot Sequoia camera includes a separate sunshine sensor that measures solar irradiance in the same spectral bands as the four image sensors. Flight plans were designed for 80% image overlap along flight paths. In addition to the radiometric corrections made by the internal solar irradiance sensor, corrections were conducted through in situ spectral measurements with black-and-white ground calibration targets, bare soil, and wheat plots using the JAZ-3 Ocean Optics STS VIS spectrometer (Ocean Optics, Inc., Dunedin, FL, USA) with a wavelength response from 350 to 800 nm (Figure 2b, Table S2). In 2017, data from white calibration targets was not used due to saturation problems (Table S2). The calibration of the spectrometer measurements was taken using a reference panel (white color Spectralon and dark) laid on the ground as targets before and after the flights. Image orthorectification was completed using ground control points (GCP). The position of the center of each GCP was acquired with a handheld GPS (Global Position-

ing System) (Geo7x, Trimble GeoExplorer series, Sunnyvale, CA, USA). All images were mosaicked using the Agisoft Photoscan Professional software (Agisoft LLC., St. Petersburg, Russia) and pixel-based georectification was done with the software QGIS version 3.2.0 (USA, http://www.qgis.org). The collected multispectral images were used to calculate several vegetation indices (VI), which were carefully selected based on the relationship to certain specific features of plant physiology [33] (Table 3).

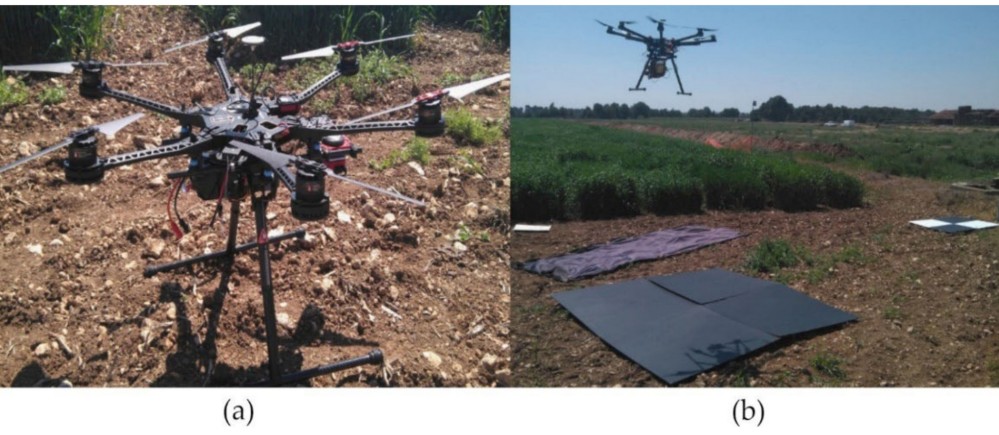

**Figure 2.** (**a**) Unmanned aerial vehicle (UAV) Hexacopter DJI S800 EVO used to collect the multispectral images of the experimental plots; (**b**) reference targets used for the geometric and radiometric calibrations.

**Table 3.** Vegetation spectral indices evaluated in this study.

| Vegetation Index | Equation | Reference |
|:---:|:---:|:---:|
| NDVI | $(R_{790} - R_{660})/(R_{790} + R_{660})$ | [33] |
| RDVI | $(R_{790} - R_{660})/\sqrt{R_{790} + R_{660}}$ | [34] |
| MSAVI | $\frac{1}{2}\left[2\,R_{790} + 1 - \sqrt{(2\,R_{790} + 1)^2 - 8\,(R_{790} - R_{660})}\right]$ | [35] |
| MTVI2 | $\dfrac{[1.5\,(1.2\,(R_{790} - R_{550}) - 2.5\,(R_{660} - R_{550}))]}{\sqrt{(2\,R_{790}+1)^2 - (6\,R_{790} - 5\,\sqrt{R_{660}}) - 0.5}}$ | [36] |
| TCARI/OSAVI | $\dfrac{3\,[(R_{735} - R_{660}) - 0.2\,(R_{735} - R_{550})\,(R_{735}/R_{660})]}{(1+0.16)\,\frac{(R_{790} - R_{660})}{(R_{790} + R_{660} + 0.16)}}$ | [37] |
| GNDVI | $(R_{790} - R_{550})/(R_{790} + R_{550})$ | [38] |

*2.3. Statistical Analysis*

Restricted maximum likelihood (REML) was used to estimate the variance components and produce the best linear unbiased predictors (BLUPs) for agronomical traits, VIs, and RGB indices, following the MIXED procedure of the SAS-STAT statistical package (SAS Institute, Inc., Cary, NC, USA). To assess differences between genotypes, years, and flight occasions, one-way ANOVAs were conducted separately for the 181 landraces and the 184 modern cultivars. LAI measurements were regressed with all the VIs described previously using aggregated data of the two growing seasons for landrace (N = 320) and modern (N = 320) panels separately and joining both panels (N = 640). Stepwise linear regression models were fit to the relationships between genotypic means for agronomic traits as dependent variables and UAV or RGB vegetation indices calculated at each flight occasion as independent ones. Since 12 landrace cultivars were considered outliers for its VI values, stepwise linear regression was conducted on 169 landraces and 184 modern cultivars. To assess the relationship between agronomic traits (yield, biomass, NSm$^2$, NGm$^2$, and TKW) and VIs, both the landrace and modern sets were randomly and equally divided into two independent groups: one for training purposes called training dataset and the other as an evaluation group for the prediction accuracy called test dataset. All the

statistical analyses and randomly splitting data for predictive modelling were carried out using the JMP v13.1.0 statistical package (SAS Institute, Inc.), considering a significance level of $p < 0.05$.

## 3. Results

### 3.1. Environmental Conditions

The experimental site is representative of the Mediterranean climate, characterized by an uneven distribution of rainfall during the season, low temperatures in winter that rise sharply in spring, and high temperatures continuing until the end of the crop cycle (Figure 3). The first growing season had less rainfall (105 mm) than the second one (269 mm) during the growth cycle from sowing (December) to maturity (June). Moreover, water scarcity was significantly higher in the 2016–2017 growing season than in the 2017–2018 growing season, mostly during the grain-filling periods, which received 5 mm and 147 mm of rainfall, respectively.

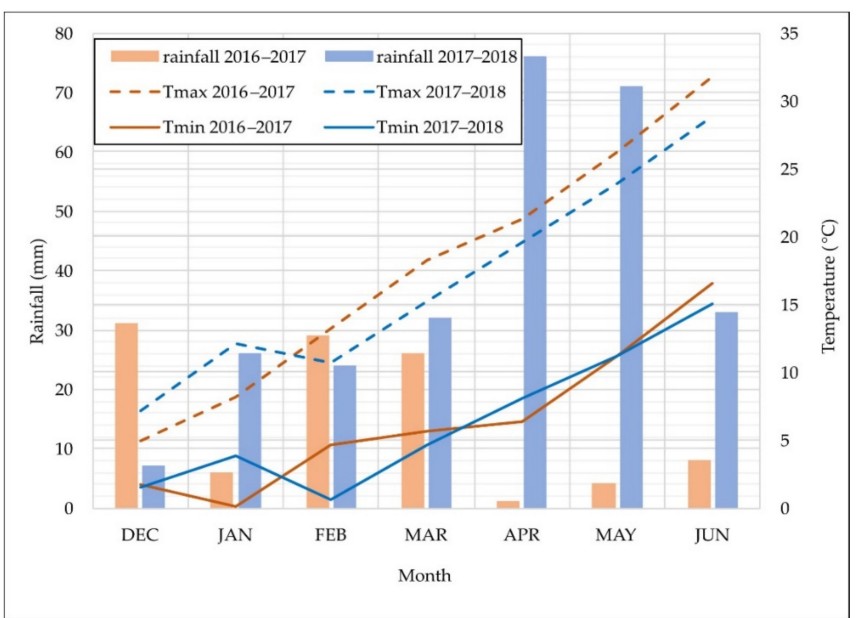

**Figure 3.** Monthly rainfall (mm), and minimum (Tmin) and maximum (Tmax) temperatures during the growth cycle of each growing season.

### 3.2. Agronomic Performance

The number of genotypes, minimum, maximum, mean, and standard deviation (SD) values for each dataset, trait, and growing season are shown in Table 4. The analysis of variance (ANOVA) for the agronomic traits was performed separately for landraces and modern genotypes (Table 5). Given that the year effect was significant for all traits in the two types of germplasm (except for biomass in the landrace set), the results are presented independently for each growing season. The percentage of variability for all traits explained by genotype was much higher than that explained by the year or by the year x genotype interaction. The contribution to total variation by the year effect was lower than that of the year x genotype interaction for all traits, except for $NGm^2$ in both landrace and modern genotypes and for TKW in modern genotypes. F-values showed that all agronomic traits, except biomass, differed significantly between landraces and modern genotypes. All the evaluated traits, except thousand kernel weight (TKW), were higher in 2018 than in 2017 in the whole collection. Grain yield, $NGm^2$, TKW, and biomass were also higher in modern cultivars than in landraces in both years. The evaluated traits had a higher coefficient of variability (CV) in both years.

**Table 4.** Main descriptive statistics for yield (t/ha), biomass at ripening (t/ha), number of spikes per square meter (NSm$^2$), number of grains per square meter (NGm$^2$), and thousand kernel weight (TKW, g) for the sample datasets used in the models. N, number of genotypes; Min, minimum values; Max, maximum values; SD, standard deviation.

| Set | Agronomic Traits | Training | | | | | Test | | | | |
|---|---|---|---|---|---|---|---|---|---|---|---|
| | | N | Min | Max | Mean | SD | N | Min | Max | Mean | SD |
| Landrace 2016–2017 | Yield (t/ha) | 84 | 3.0 | 8.3 | 5.0 | 0.9 | 85 | 3.2 | 8.5 | 5.2 | 0.9 |
| | Biomass (t/ha) | | 8.6 | 24.5 | 15.5 | 3.5 | | 6.5 | 24.5 | 15.9 | 3.4 |
| | NSm$^2$ | | 386 | 761 | 544 | 73 | | 381 | 686 | 542 | 63 |
| | NGm$^2$ | | 7438 | 20,154 | 12,764 | 2481 | | 8180 | 23,003 | 13,082 | 2742 |
| | TKW (g) | | 27.0 | 51.6 | 38.9 | 5.2 | | 23.3 | 52.7 | 39.8 | 5.1 |
| Landrace 2017–2018 | Yield (t/ha) | 84 | 3.6 | 7.2 | 5.5 | 0.7 | 85 | 4.1 | 9.0 | 5.8 | 0.9 |
| | Biomass (t/ha) | | 7.1 | 29.9 | 16.1 | 4.7 | | 6.7 | 33.9 | 16.9 | 5.2 |
| | NSm$^2$ | | 372 | 824 | 580 | 94 | | 345 | 889 | 583 | 95 |
| | NGm$^2$ | | 13,035 | 22,227 | 16,357 | 1645 | | 12,917 | 24,836 | 17,130 | 2650 |
| | TKW (g) | | 19.9 | 49.3 | 33.9 | 6.3 | | 17.9 | 49.0 | 34.6 | 7.5 |
| Modern 2016–2017 | Yield (t/ha) | 92 | 7.1 | 11.8 | 9.5 | 0.9 | 92 | 6.5 | 11.7 | 9.4 | 1.1 |
| | Biomass (t/ha) | | 8.5 | 22.9 | 16.4 | 2.9 | | 10.2 | 22.9 | 16.3 | 3.0 |
| | NSm$^2$ | | 253 | 820 | 486 | 117 | | 280 | 813 | 471 | 108 |
| | NGm$^2$ | | 14,276 | 31,452 | 22,630 | 3628 | | 12,520 | 33,852 | 22,170 | 4251 |
| | TKW (g) | | 31.3 | 58.8 | 42.8 | 5.2 | | 32.6 | 58.1 | 43.3 | 5.1 |
| Modern 2017–2018 | Yield (t/ha) | 92 | 6.9 | 12.0 | 10.0 | 1.0 | 92 | 7.3 | 12.4 | 10.0 | 1.0 |
| | Biomass (t(ha) | | 10.4 | 39.0 | 19.2 | 4.6 | | 6.2 | 29.4 | 19.6 | 3.9 |
| | NSm$^2$ | | 200 | 973 | 583 | 149 | | 220 | 920 | 585 | 142 |
| | NGm$^2$ | | 17,002 | 34,191 | 26,848 | 3706 | | 17,752 | 41,629 | 26,718 | 4082 |
| | TKW (g) | | 29.7 | 51.1 | 37.7 | 4.2 | | 24.4 | 51.3 | 38.1 | 4.6 |

**Table 5.** Analysis of variance performed separately for 181 landraces and 184 modern cultivars and values for grain yield, biomass, number of spikes per square meter (NSm$^2$), number of grains per square meter (NGm$^2$), and thousand kernel weight (TKW) for each growing season. SS, sum of squares. CV, coefficient of variability. ** $p < 0.01$. *** $p < 0.001$.

| | | Landraces | | | | | Modern | | | | |
|---|---|---|---|---|---|---|---|---|---|---|---|
| | | Yield (t/ha) | Biomass (t/ha) | NSm$^2$ | NGm$^2$ | TKW (g) | Yield (t/ha) | Biomass (t/ha) | NSm$^2$ | NGm$^2$ | TKW (g) |
| SS Year (%) | | 8.4 | 0.8 | 5.3 | 38.3 | 15.0 | 6.2 | 14.2 | 14.2 | 23.9 | 22.5 |
| SS Genotype (%) | | 63.7 | 52.6 | 55.9 | 40.1 | 64.8 | 64.7 | 42.9 | 50.6 | 62.5 | 64.2 |
| SS Year × Genotype (%) | | 27.9 | 46.6 | 38.8 | 21.5 | 20.1 | 29.1 | 42.9 | 35.2 | 13.6 | 13.3 |
| F year | | 50.3 *** | 2.8 | 23.1 *** | 296.7 *** | 125.6 *** | 38.8 *** | 61.0 *** | 74.0 *** | 322.1 *** | 309.2 *** |
| F genotype | | 2.3 *** | 1.1 | 1.4 ** | 1.8 *** | 3.2 *** | 2.2 *** | 1.0 | 1.4 ** | 4.6 *** | 4.8 *** |
| CV (%) | 2016–2017 | 17.9 | 22.2 | 12.5 | 20.2 | 13.1 | 10.7 | 18.0 | 23.5 | 17.6 | 11.9 |
| | 2017–2018 | 14.7 | 30.1 | 16.2 | 13.4 | 20.1 | 9.9 | 22.1 | 24.9 | 14.5 | 11.6 |
| Mean | 2016–2017 | 5.1 | 15.7 | 543 | 12,923 | 39.4 | 9.5 | 16.4 | 479 | 22,400 | 43.0 |
| | 2017–2018 | 5.6 | 16.5 | 582 | 16,746 | 34.3 | 10.0 | 19.4 | 584 | 26,783 | 37.9 |
| Minimum | 2016–2017 | 3.0 | 6.5 | 381 | 7438 | 23.3 | 6.5 | 8.5 | 253 | 12,520 | 31.3 |
| | 2017–2018 | 3.6 | 6.7 | 345 | 12,917 | 17.9 | 6.9 | 6.2 | 200 | 17,002 | 24.4 |
| Maximum | 2016–2017 | 8.5 | 24.5 | 761 | 23,003 | 52.7 | 11.8 | 22.9 | 820 | 33,852 | 58.8 |
| | 2017–2018 | 9.0 | 33.9 | 889 | 24,835 | 49.3 | 12.4 | 39.0 | 973 | 41,629 | 51.3 |

*3.3. LAI Prediction through Vegetation Indices*

Estimates of LAI were carried out with aggregated data of the two growing seasons for landraces and modern sets separately. Although LAI measurements were regressed with all the VIs reported in Tables 2 and 3, only the NDVI, GNDVI, modified triangular vegetation index (MTVI2), GA, GGA, Hue, a*, and u* showed significant relationships ($p < 0.001$) (Table 6). Despite the lower R$^2$ values for landraces, LAI predictions for both panels showed similar slopes for the relation between observed LAI and estimated LAI. Thus, LAI was assessed for the whole collection, joining data from landraces and modern

genotypes of the two growing seasons. The highest $R^2$ for LAI estimates using UAV multispectral images was obtained with the MTVI2 ($R^2$ = 0.61), which showed a RMSE of 1.17. On the other hand, Hue was the ground-based RGB index with the highest $R^2$ ($R^2$ = 0.45) and a RMSE of 1.40.

**Table 6.** Statistically significant (*p* < 0.001) relationships between leaf area index (LAI) measured with the ceptometer and vegetation indices (VIs) obtained from UAV multispectral and RGB images. Calculations have been made with aggregated data of the two growing seasons and image acquisition occasions joining germplasm collections and for landraces and modern sets separately. ns, no significant. RMSE, root mean square error.

| Method | VI | Equation | $R^2$ | RMSE | Equation | $R^2$ | RMSE | Equation | $R^2$ | RMSE |
|---|---|---|---|---|---|---|---|---|---|---|
| | | *Landraces + Modern* (N = 640) | | | *Landraces* (N = 320) | | | *Modern* (N = 320) | | |
| UAV Multispectral | NDVI | y = 11.63x − 5.55 | 0.38 | 1.48 | y = 10.74x − 4.49 | 0.16 | 1.45 | y = 11.06x − 5.22 | 0.43 | 1.47 |
| | GNDVI | y = 8.89x − 2.54 | 0.18 | 1.70 | ns | | | y = 9.42x − 3.35 | 0.26 | 1.67 |
| | MTVI2 | y = 7.45x − 1.01 | 0.61 | 1.17 | y = 7.11x − 0.72 | 0.39 | 1.24 | y = 7.58x − 1.10 | 0.66 | 1.12 |
| Ground-based RGB | GA | y = 7.18x − 1.36 | 0.41 | 1.43 | y = 8.04x − 2.00 | 0.20 | 1.41 | y = 6.71x − 1.09 | 0.45 | 1.43 |
| | GGA | y = 4.52x − 1.86 | 0.39 | 1.45 | y = 4.47x + 2.07 | 0.29 | 1.33 | y = 4.27x + 1.89 | 0.38 | 1.52 |
| | Hue | y = 0.09x − 2.91 | 0.45 | 1.40 | y = 0.10x − 3.59 | 0.33 | 1.29 | y = 0.08x − 2.44 | 0.45 | 1.44 |
| | a* | y = 0.18x − 2.04 | 0.22 | 1.66 | ns | | | y = −0.18x + 1.96 | 0.21 | 1.73 |
| | u* | y = 0.19x − 3.05 | 0.3 | 1.57 | y = -0.16x + 3.54 | 0.15 | 1.46 | y = −0.18x + 2.99 | 0.3 | 1.63 |

Then, the LAI of all plots was estimated through MTVI2, considering the growth stage of each genotype at each flight occasion. LAI varied significantly between the set of genotypes and years (*p* < 0.001) at each flight occasion and growth stage. Figure 4 shows that LAI was higher in 2018 than in 2017 for both landraces and modern cultivars. In the first growing season (2016–2017), landraces had LAI values significantly higher than those of modern cultivars at 128 DAS and 151–153 DAS, but similar at 178–184 DAS (Figure 4a). Maximum LAI values for landraces and modern cultivars in 2017 were obtained at the booting and stem elongation stages, respectively (Figure 4b). The LAI of landraces in 2017 was significantly higher than modern cultivars until anthesis, when it decreased significantly until the values were lower than those estimated in the modern panel. Therefore, the LAI of modern cultivars started declining later than in landraces. In 2018, the LAI of landraces and modern cultivars had a similar pattern throughout the growing season without significant differences between them, except at the hard dough-grain stage, where the LAI of landraces was slightly lower than that of modern cultivars (Figure 4b).

### 3.4. Performance of Stepwise Regression Models

Table 7 shows the main statistics of the models built to estimate the different agronomic traits with UAV multispectral and RGB VIs for each year and germplasm set. Scatter plots for the relation between estimated and observed agronomic traits on the test dataset based on Table 7 equations are shown in Figures S1 and S2. The results indicate that the training models developed from multispectral images were significant for all traits, germplasm sets, and years, with the exception of NSm$^2$ for the landraces set in 2018. Grain yield and NGm$^2$ were the traits showing the highest $R^2$ in both germplasm collections. Most of the equations developed with multispectral VI had in common the NDVI and GNDVI indices, although in some cases MTVI and MSAVI also appeared. The models constructed with RGB-VI were also statistically significant in all cases except for biomass and NSm$^2$ for the landraces set in 2018. Yield was also one of the most predictive traits.

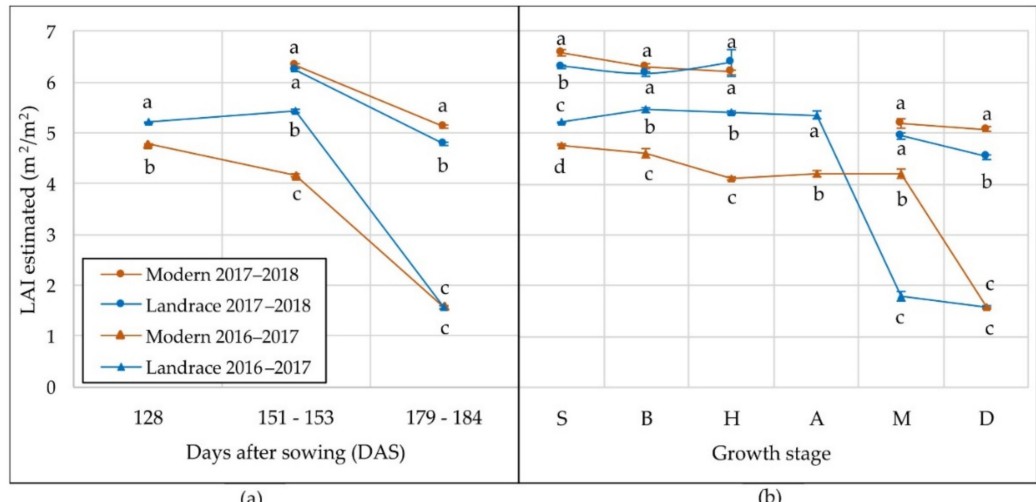

**Figure 4.** Mean values in 2017 and 2018 of leaf area index estimated through MTVI2 for landraces and modern cultivars at: (**a**) each date of image acquisition expressed in days after sowing (DAS), and (**b**) each growth stage. S, stem elongation; B, booting; H, heading; A, anthesis; M, milk-grain development; D, hard dough-grain development. Different letters at each date or growth stage indicate significant differences at $p \leq 0.01$ using Tukey's honest significant difference test.

Test models obtained with the corresponding dataset also showed the highest $R^2$ for yield and $NGm^2$, using either multispectral or RGB VIs. However, $R^2$ tended to be slightly lower for the latter. For multispectral VI, the maximum $R^2$ obtained to predict yield in landraces and modern cultivars was 0.36 and 0.43, respectively, which corresponded to an RMSE of 0.28 t/ha and 0.39 t/ha. In addition, the maximum $R^2$ for $NGm^2$ predictions through multispectral VIs was 0.19 and 0.38 for landraces and modern genotypes, respectively, corresponding with RMSE values of 768 and 1835 grains/$m^2$ (Table 7). Considering training and test model values together, the highest $R^2$ for yield was obtained in modern genotypes, being higher for the growing season 2016-2017 ($R^2 = 0.43$ and $R^2 = 0.37$ through UAV and RGB imagery, respectively) than in the next growing season ($R^2 = 0.29$ and $R^2 = 0.45$ through UAV and RGB imagery, respectively).

Table 8 shows the training and test statistics for the five agronomic traits obtained with aggregated datasets of the two growing seasons for landraces and modern cultivars. Scatter plots for the relation between estimated and observed agronomic traits on the test dataset based on Table 8 equations are shown in Figures S3 and S4. In general, the models fitted better for modern cultivars. The test models for most agronomic traits were not significant in the set of landraces. In general, both test and training models improved when the data from two growing seasons were analyzed together. Grain yield and $NGm^2$ were again the traits that showed the highest $R^2$, using either UAV multispectral and RGB VIs (Table 8). For these two traits, despite the $R^2$ of training models being higher in modern cultivars, the RMSE tended to be lower in landraces. For the models built with multispectral VI, the RMSE in yield predictions ranged from 0.26 to 0.32 t/ha and from 0.34 to 0.38 t/ha, for landraces and modern cultivars, respectively. The models built with ground-based RGB VIs had RMSE values ranging from 0.28–0.50 t/ha and 0.39–0.54 t/ha for landraces and modern cultivars, respectively. The highest $R^2$ for yield training models of landraces were obtained with ground-based RGB VI, testing data with the dataset corresponding to 2018 ($R^2 = 0.30$). In contrast, training model of yield in modern genotypes had the highest $R^2$ using UAV multispectral VI, testing data in the dataset of 2017 ($R^2 = 0.51$).

**Table 7.** Training and test statistics of the models for the estimations of agronomic traits through UAV multispectral and RGB VIs for each germplasm set and growing season. * $p < 0.05$. ** $p < 0.01$. N, number of genotypes; $R^2$, determination coefficient; RMSE, root mean standard error; Yield (t/ha); $NSm^2$, number of spikes per square meter; $NGm^2$, number of grains per square meter; TKW, thousand kernel weight (g); I, intensity; L, lightness; S, saturation. Number after each VI means the flight occasion: 1, 128 DAS; 2, 151-153 DAS; 3, 179-184 DAS.

| | | UAV Multispectral | | | | | | Ground-Based RGB | | | | | |
|---|---|---|---|---|---|---|---|---|---|---|---|---|---|
| | | Training | | | Test | | | Training | | | Test | | |
| Set | Traits | N | Equation | $R^2$ | N | $R^2$ | RMSE | N | Equation | $R^2$ | N | $R^2$ | RMSE |
| Landraces 2016–2017 | Yield | 84 | $-26.69 + 31.89GNDVI_1 + 5.98NDVI_3$ | 0.18 ** | 85 | 0.18 ** | 0.45 | 84 | $40.45 + 2.26GA_3 + 12.77S_3 - 0.91L_1 - 0.15b^*_2$ | 0.45 ** | 85 | 0.28 ** | 0.66 |
| | Biomass | | $-130.27 + 29.05MSAVI_2 + 135.54GNDVI_2$ | 0.18 ** | | ns | 1.41 | | $-17.65 + 40.58GGA_1$ | 0.11 ** | | ns | 1.04 |
| | $NSm^2$ | | $-2167.28 + 2958.35GNDVI_2$ | 0.15 * | | ns | 27.22 | | $-731.73 + 8062.69I_3 - 58.49L_3 - 64.60a^*_2$ | 0.24 ** | | 0.10 ** | 37.05 |
| | $NGm^2$ | | $-73937 + 9059.90NDVI_3 + 92920GNDVI_1$ | 0.27 ** | | 0.08 ** | 1204 | | $34,836 - 544.90L_2$ | 0.12 ** | | ns | 867 |
| | TKW | | $172.87 - 173.71GNDVI_2 + 40.10GNDVI_3$ | 0.15** | | ns | 1.86 | | $-115.24 + 614.25I_2 - 106.53I_3$ | 0.17** | | ns | 2.06 |
| Landraces 2017–2018 | Yield | 84 | $0.006 + 7.01GNDVI_3$ | 0.18 ** | 85 | 0.36 ** | 0.28 | 84 | $-4.53 + 0.08Hue_2 - 0.18a^*_3$ | 0.33 ** | 85 | 0.25 ** | 0.34 |
| | Biomass | | $-26.24 + 46.70MSAVI_2$ | 0.10 ** | | ns | 1.34 | | ns | ns | | ns | ns |
| | $NSm^2$ | | ns | ns | | ns | ns | | ns | ns | | ns | ns |
| | $NGm^2$ | | $-2,167,724 + 2,631,346GNDVI_2 + 12,018GNDVI_3$ | 0.24 ** | | 0.19 ** | 768 | | $19,077 - 25,561I_2 - 453.53a^*_3$ | 0.10 ** | | 0.16 ** | 590 |
| | TKW | | $-11.97 + 57.72RDVI_2$ | 0.15 ** | | ns | 2.22 | | $52.23 - 209.47S_2 - 1.72a^*_2$ | 0.29 ** | | 0.11 ** | 2.92 |
| Modern 2016–2017 | Yield | 92 | $-9.23 + 8.09NDVI_3 + 19.68MTVI_2$ | 0.28 ** | 92 | 0.43 ** | 0.39 | 92 | $4.93 - 0.15a^*_3 - 0.27u^*_1$ | 0.34 ** | 92 | 0.37 ** | 0.49 |
| | Biomass | | $-63.23 + 21.86MSAVI_3 + 102.20MTVI_2$ | 0.28 ** | | 0.11 ** | 1.48 | | $0.28 + 0.20Hue_2$ | 0.24 ** | | 0.22 ** | 1.14 |
| | $NSm^2$ | | $-13857 + 11,036TCARI/OSAVI_2 + 15,315GNDVI_2$ | 0.22 ** | | 0.16 ** | 56.49 | | $916.14 - 33.34L_1 - 44.65a^*_1$ | 0.21 ** | | 0.18 ** | 55.83 |
| | $NGm^2$ | | $-221,638 + 272,852GNDVI_2$ | 0.33 ** | | 0.35 ** | 1863 | | $3,4087 + 5840.71GGA_3 - 9,9931I_1 - 1085.73a^*_2$ | 0.45 ** | | 0.45 ** | 1780 |
| | TKW | | $292.96 - 279.44GNDVI_2$ | 0.17 ** | | 0.11 ** | 2.23 | | $-11.24 + 26.68GA_2 + 163.80I_1 + 2.27a^*_2 + 0.59v^*_3$ | 0.36 ** | | 0.11 ** | 5.17 |
| Modern 2017–2018 | Yield | 92 | $-13 + 25.25NDVI_3$ | 0.29 ** | 92 | 0.24 ** | 0.38 | 92 | $-1.04 - 16.13GA_2 + 9.05GA_3 + 0.19Hue_2$ | 0.45 ** | 92 | 0.22 ** | 0.54 |
| | biomass | | $-143.78 + 177.75MSAVI_2$ | 0.07 ** | | ns | 1.28 | | $-21.58 - 55.66I_3 + 0.59Hue_2$ | 0.12 ** | | ns | 1.84 |
| | $NSm^2$ | | $10,710 - 14,953MTVI2_3 + 2604GNDVI_2$ | 0.22 ** | | 0.08 ** | 72.94 | | $-594.57 - 84.23u^*_2$ | 0.06 * | | 0.06 * | 37.29 |
| | $NGm^2$ | | $-52,520 + 34,921MSAVI_3 + 59,546GNDVI_2$ | 0.49 ** | | 0.38 ** | 1835 | | $34,497 - 447.30L_3 - 2198.99u^*_3$ | 0.32 ** | | 0.21 ** | 1825 |
| | TKW | | $82.24 - 52.04GNDVI_2$ | 0.15 ** | | 0.10 ** | 1.55 | | $57.85 - 0.87b^*_2$ | 0.14 ** | | 0.04 * | 1.34 |

**Table 8.** Training and Test statistics of the models for the estimations of agronomic traits through UAV multispectral and RGB VIs aggregating the data of the two growing seasons for landraces and modern cultivars. * $p < 0.05$. ** $p < 0.01$. N, number of genotypes; $R^2$, determination coefficient; RMSE, root mean standard error; Yield (t/ha); $NSm^2$, number of spikes per square meter; $NGm^2$, number of grains per square meter; TKW, thousand kernel weight (g); I, intensity; L, lightness; S, saturation. Number after each VI means the flight occasion: 1, 128 DAS; 2, 151-153 DAS; 3, 179-184 DAS.

| Set | Traits | Training | | | Test 2016–2017 | | | Test 2017–2018 | | | Test 2016–2017+2017–2018 | | |
|---|---|---|---|---|---|---|---|---|---|---|---|---|---|
| | | **UAV Multispectral** | | | | | | | | | | | |
| | | N | Equation | $R^2$ | N | $R^2$ | RMSE | N | $R^2$ | RMSE | N | $R^2$ | RMSE |
| Landraces 2016–2017+2017–2018 | Yield | 168 | $0.30 + 11.26NDVI\_3 - 5.11MTVI2\_3$ | 0.25 ** | 85 | 0.17 ** | 0.27 | 85 | 0.27 ** | 0.26 | 170 | 0.28 ** | 0.32 |
| | Biomass | | $-50.03 + 35.23MTVI2\_2 + 38.30GNDVI\_2$ | 0.11 ** | | ns | - | | ns | - | | ns | - |
| | $NSm^2$ | | ns | ns | | ns | - | | ns | - | | ns | - |
| | $NGm^2$ | | $-713.21 + 21637GNDVI\_3$ | 0.44 ** | | ns | 545 | | 0.17 ** | 980.73 | | 0.42 ** | 1376 |
| | TKW | | $-80.01 + 29.67RDVI\_2 + 109.65GNDVI\_2$ | 0.16 ** | | ns | 0.99 | | ns | - | | 0.14 ** | 2.20 |
| Modern 2016–2017+2017–2018 | Yield | 184 | $-5.19 + 9.47GNDVI\_2 + 13.63NDVI\_3 - 6.46MTVI2\_3$ | 0.30 ** | 92 | 0.51 ** | 0.34 | 92 | 0.33 ** | 0.38 | 184 | 0.46 ** | 0.38 |
| | Biomass | | $-63.23 + 102.20MTVI2\_2 + 21.86MSAVI\_3$ | 0.28 ** | | 0.11 ** | 1.48 | | 0.01 | 2.38 | | 0.20 ** | 17.73 |
| | $NSm^2$ | | $-1906.39 + 1047.64NDVI\_2 + 1265.34GNDVI\_2 + 477.01GNDVI\_3$ | 0.27 ** | | 0.24 ** | 46.45 | | 0.10 ** | 55.94 | | 0.29 ** | 60.88 |
| | $NGm^2$ | | $-69278 + 20,340NDVI\_2 + 22,033NDVI\_3 + 66,300GNDVI\_2$ | 0.54 ** | | 0.49 ** | 1419 | | 0.32 ** | 2058.47 | | 0.53 ** | 2091 |
| | TKW | | $92.05 - 37.35GNDVI\_2 - 26.21GNDVI\_3$ | 0.31** | | 0.19 ** | 1.13 | | 0.10 ** | 1.67 | | 0.33 ** | 2.42 |
| | | **Ground-based RGB** | | | | | | | | | | | |
| Landraces 2016–2017+2017–2018 | Yield | 168 | $3.91 - 0.07L\_2 - 0.10a*\_2 - 0.14b*\_2 + 0.05Hue\_3 + 9.19S\_3$ | 0.36 ** | 85 | 0.21 ** | 0.50 | 85 | 0.30 ** | 0.28 | 170 | 0.28 ** | 0.42 |
| | biomass | | ns | ns | | ns | - | | ns | - | | ns | - |
| | $NSm^2$ | | $412.15 - 13.54u*\_2 - 84.37GGA\_3$ | 0.09 ** | | ns | - | | ns | - | | 0.09 ** | 24.31 |
| | $NGm^2$ | | $25,069 + 6675.08GGA\_2 - 34,739I\_2 - 292.79L\_2 + 11,005GA\_3 + 426.02u*\_3$ | 0.50 ** | | ns | - | | 0.14 ** | 804.53 | | 0.39 ** | 1604 |
| | TKW | | $32.37 - 27.23GGA\_2 - 3.93b*\_2 + 4.06v*\_2 + 0.13Hue\_3$ | 0.28 ** | | ns | 1.94 | | 0.04 | 2.50 | | 0.15 ** | 2.98 |
| Modern 2016–2017+2017–2018 | Yield | 184 | $10.64 - 4.95GA\_3 - 7.68I\_3 + 0.07Hue\_3 - 0.22u*\_3$ | 0.28 ** | 92 | 0.27 ** | 0.51 | 92 | 0.24 ** | 0.39 | 184 | 0.30 ** | 0.54 |
| | Biomass | | $-0.28 + 0.20Hue\_2$ | 0.24 ** | | 0.09 ** | 1.90 | | ns | - | | 0.20** | 2.09 |
| | $NSm^2$ | | $238.83 - 18.41a*\_2$ | 0.19 ** | | 0.20 ** | 30.24 | | 0.05* | 30.87 | | 0.26 ** | 50.66 |
| | $NGm^2$ | | $21,688 - 37,911I\_3 + 143.07Hue\_3 - 373.30a*\_2$ | 0.45 ** | | 0.45 ** | 1314 | | 0.19 ** | 1257.45 | | 0.45 ** | 2221 |
| | TKW | | $24.78 + 39.88GA\_2 + 0.63a*\_2 + 1.04u*\_2$ | 0.36** | | 0.28 ** | 1.45 | | 0.11 ** | 1.31 | | 0.36 ** | 2.37 |

## 4. Discussion

The current study evaluates the suitability of using a 4-band multispectral camera (Parrot Sequoia) on-board UAV and ground-based RGB images to predict yield in wheat under a rainfed Mediterranean-type environment. Despite remote sensing methods being nondestructive and cost-efficient approaches based on the information provided by visible and near-infrared (VIS-NIR) radiation reflection [39], care should be taken to standardize measurements across different plant architectures and sun elevation [6]. The light intensity, temperature, cloud cover, wind speed, and timing of measurements can also affect the accuracy of the estimation of traits evaluated in the field [40]. Digital photography is also a promising approach due to the use of conventional cameras as a low-cost sensor to get the image and open-source software to process the data from it [25].

The large year effect for the assessed traits found in the current study may be attributed to the contrasting water availability in the two years of the experimental fields, which doubled in 2018 compared to the preceding year. The largest differences were observed in April and beyond, coinciding with the grain-filling period, which likely was the main cause of the lower yield, spike number, and grain number recorded in 2017 compared with 2018. It is well known that water scarcity after anthesis has significant effects on yield and yield components [12,33,41]. The heaviest kernels observed in 2017 were most probably a consequence of the compensation between yield components, since a lower NGm$^2$ was observed in 2017. It has been shown that the value of each component strongly depends on the values of the components defined previously, and NGm$^2$ is defined before TKW [42]. The number of grains and their weight are established sequentially during plant development, with the potential number of grains being determined before anthesis, and the grain weight being fixed after it [42,43]. This is in accordance with the heaviest grains being obtained in the current study in 2017, the year with the lowest grain number. The high yields achieved in the two years are in agreement with those reported in previous studies at the same experimental site [44], where the high yields could be attributed to the high soil fertility (about 3% organic matter) and the superficial subsoil water layer at this site [45]. The CVs obtained in the current study for the analyzed traits are within the normal ranges reported for water-limited environments [10]. Moreover, the largest variability of agronomic traits found in landraces when compared with modern varieties is in agreement with the results of previous studies conducted in durum wheat [41,46].

The remotely sensed estimates of LAI in both landraces and modern cultivars were higher in 2018 (Figure 3), as well as grain yield, which may be mostly explained by the higher rainfall received during the grain-filling period in that year. As reported by Villegas et al. [47], drought severely affects the total above-ground biomass due to a decrease in the rate of growth. Although water stress affects the growth of wheat, the effects are less harmful at the early stages of the crop cycle than during the grain-filling period [48]. In 2018, the LAI of the landrace was slightly lower than that of the modern cultivar only at the end of the growing season (the dough development stage), suggesting that, under well-watered conditions, the vegetative growth capacity of the latter is higher or senescence of the former starts earlier. Villegas et al. [42] and Royo et al. [49] reported similar conclusions. In 2017, however, the LAI of landraces was significantly higher than that of modern genotypes until anthesis, which could be due to the higher resistance to water stress of landraces [42,50], their superior water use efficiency before flowering [51], and their large root system [52], which is able to exploit deeper soil layers. Figure 3 shows that, despite the LAI of landraces in 2017 being higher than that of modern genotypes until anthesis, it started declining earlier than the latter. This anticipated decrease of LAI in the landrace genotypes could be partially explained by a higher water demand of landraces, a consequence of their larger canopy, which could not be fulfilled at the end of the growing season, leading to the anticipated senescence. Moreover, it could be partially attributed to the greater potential of modern cultivars, compared with the landraces, to use water during grain filling to achieve yield increases [51]. It is also important to mention that differences in remotely sensed estimates of LAI between phenological stages could also be influenced by differences in the

chlorophyll content. It is well known that chlorophyll content decreases during senescence and as a consequence, also those VIs that uses bands mostly placed in the NIR and green regions [53]. Therefore, it may happen that plants with the same LAI at different growing stages had different value of a VI due to differences in the chlorophyll content. Despite of this, Din et al. [54] reported that the MTVI2 was one of the most consistent VIs to change through phenological stages. However, it is possible that the estimates of the low LAI values at the end of the growing season could also be affected by a low chlorophyll content due to senescence, as previously mentioned.

A number of studies have estimated agronomic traits such as grain yield or biomass through UAV multispectral and RGB imagery in wheat and other cereals, but the majority of them have been conducted in irrigated environments [9,16,55] or under a wide range of phenotypic variability resulting from varying growing conditions [9,56,57]. A proper assessment of agronomic traits through remote sensing is expected when phenotypic variability is present. This usually occurs in experiments conducted under irrigated conditions, where genotypes are allowed to express their full potential, thus, maximizing differences between them [17], or when a wide range of phenotypic values results from treatments varying the agronomic management [9,56,57]. However, studies conducted in wheat under rainfed conditions are scarce and the precision of the assessments obtained on them is lower. A study by Kyratzis et al. [12], conducted on durum wheat, obtained $R^2$ values of $\leq 0.43$ for the relationships between NDVI and yield at different growth stages, which are comparable to the values reported here.

In this study, MTVI2 was the best VI to estimate LAI through multispectral imagery ($R^2 = 0.61$). On the other hand, estimates of LAI through RGB VIs showed slightly lower $R^2$, with Hue being the best predictor ($R^2 = 0.45$). It is widely known that some vegetation indices, such as NDVI, show saturation when LAI reports high values [12,36,56]. Furthermore, estimating green LAI through the NDVI has several limitations since it is affected, for instance, by factors such as soil background, canopy shadows, atmospheric conditions, and variations in leaf chlorophyll concentration [58]. Haboudane et al. [36] stated that improved VIs such as MTVI were more sensitive to chlorophyll variations and, therefore, responded better to LAI changes. In addition, it has been reported that MTVI2 is better than other VIs mitigating this saturation effect in wheat with LAI values ranging from 2 to 8 [54,56,59]. Despite LAI was obtained through destructive measurements, results from our study had similar LAI values and the regression with the MTVI2 showed a RMSE of 1.17. This RMSE agrees with values obtained by Xing et al. [59], who reported RMSE values ranging from 1.1 to 1.6 when using different VIs calculated with a spectrometer and Sentinel 2 imagery. In particular, the RMSE of MTVI2 obtained by these authors was 1.26 and 1.16 using the spectrometer and Sentinel 2 imagery, respectively, which agreed with the RMSE obtained in our study. Hassan et al. [60] also exhibited a strong relationship between VIs and LAI measured with an AccuPAR LP-80 ceptometer with values ranging from 2 to 5.5.

The current study demonstrated that predictions of yield could be properly obtained using both multispectral and RGB VI, with the $R^2$ of the latter tending to be higher. Although models differed depending on the type of germplasm and the trait to be assessed, NDVI and GNDVI were the VIs mostly entered in all of the prediction equations obtained through UAV multispectral imagery, thus, confirming the feasibility of using such structural VIs to assess different agronomic traits in wheat [14,17,39,61]. On the other hand, as mentioned above, ground-based RGB imagery showed better estimations than UAV multispectral imagery for the prediction set. RGB indices such as GA, GGA, a*, and u* have been proven to be more suitable for predicting higher yield due to their capacity to calculate a combination of physiological components related to biomass [25,26]. Kefauver et al. [27] and Gracia-Romero et al. [9] reported the feasibility of using RGB VI to estimate different agronomic traits. In this study, a positive and negative contribution of GA and a*, respectively, at the last image acquisition date (DAS 179-184) were present in most of the algorithms for predicting yield. This confirms that the indices that performed better

in assessing differences in yield were the ones related to canopy greenness and, thus, to vegetation cover [62]. GA quantifies the green pixels of the total pixels in the image, and, thus, is reliable to use for estimating the fraction of vegetation cover [63]. As most of the carbohydrates for grain filling are formed after heading, a larger leaf area or vegetation cover is positively correlated with grain yield, determining the future number of grains and their weight [14,25]. Accordingly, a* and u* measurements are also related to 'greenness', where the values go from high negative (green) to low negative or even positive values (lack of green). Furthermore, Rezzouk et al. [64] observed that ground-based RGB imagery presented a higher resolution than aerial images, since they found that the number of pixels per plot decreased drastically when acquiring images aerially. In our case, this was probably not the case since the pixel resolution of RGB and UAV multispectral imagery were <1 cm and 5 cm, respectively. In addition, the use of relatively low-cost RGB sensors could be a feasible alternative to multispectral cameras from UAV measurements for plant phenotyping [57].

The lower $R^2$ observed between VI and yield in landraces than in modern cultivars when the data of each year were analyzed separately could be partially due to the different size and structure of the canopy of both types of germplasm, as landraces were much taller and had a different canopy architecture, which probably saturated the VI at high LAI values. However, in all cases the $R^2$ values were $\geq 0.22$. GNDVI and NDVI were the VIs entered into the equations to estimate yield, showing in all cases positive relationships with it. This is in agreement with previous studies showing positive correlations between yield and VI in different environments [65,66], as negative relationships are more frequent under severe water stress conditions [67,68]. Yield predictions in modern genotypes through UAV multispectral VIs varied between the training and test datasets, mostly for the growing season 2016–2017. The $R^2$ of the later was slightly higher, suggesting that the model is able to improve yield predictions on dry years. This could be explained because during years with water scarcity, the variability between genotypes in traits related to leaf biochemical properties or canopy structural attributes, which can explain a part of the yield, could be higher. Biomass and the number of spikes per unit area could not be assessed in landraces in a reliable way as, although some models were statistically significant, they accounted for a small fraction of the observed variability. However, in modern cultivars predictions of biomass were year-dependent as models accounted from 11% to 28% of the observed variation in 2017 but $\leq 12\%$ in 2018. This could be due to the saturation of VI when LAI > 5, which was the case in the two germplasm sets in 2018 and in the landraces in 2017, as shown in Figure 3b. Despite this, the significant predictions of biomass were always obtained through the MSAVI index, which seeks to address some of the limitations of NDVI when applied to areas with a high degree of exposed soil surface. It was not surprising that the number of spikes per unit area could not be properly estimated through VI, as the reflectance of the spikes probably caused some distortion in measurements made in the visible and near-infrared ranges, as demonstrated in previous studies [69]. Estimations of $NSm^2$ with RGB indices were not properly assessed. The number of grains per unit area was better estimated in modern cultivars than in landraces, with both VI and RGB indices. This was not surprising given the strong relationship between the number of grains per unit area and yield in semidwarf cultivars [70,71]. Again, the relationships between VI and RGB indices with grain weight were more consistent in modern cultivars than in landraces. The negative correlations between grain weight and GNDVI revealed by some prediction models suggest that the plants with higher biomass produced lighter grains, likely as a consequence of competition between plants for allocating photosynthates in vegetative and productive structures during grain filling.

Repeated measurements of the whole collection acquired on different dates throughout the growing season are necessary to improve the prediction of agronomic traits [61]. According to this, in our study, predictions of agronomic traits improved when information from different flights was analyzed together (Table 8). The highest $R^2$ for grain yield predictions ($R^2 = 0.51$) was obtained for modern genotypes in 2017 using combined data

from flights acquired on DAS 151 (heading, anthesis, and milk development) and 179 (dough development). Despite $R^2$ being slightly lower in 2018, in all cases the RMSE varied between 0.26 and 0.38 t/ha, which demonstrates the suitability of the models developed. It has been proven in several studies [9,33,72] that higher variability within a population can increase the determination coefficient and, therefore, the predictive ability of the model.

## 5. Conclusions

The efficiency of breeding programs and the agronomic research will increase considerably depending on the reliability of models for HTP. This study demonstrated the potential of a 4-band multispectral camera (Parrot Sequoia) and RGB images for assessing agronomic traits—particularly yield and grain number per unit area—in bread wheat grown in a Mediterranean-type environment. However, the suitability of the models proved to be specific, as their consistency depended on the canopy structure, leaf dimensions and orientation, and the environmental conditions during vegetative growth, which poses a difficulty for their general use in a random crop season. Thus, uniformity in the crop cycle among cultivars seems to be essential to improve prediction models minimizing environmental effects. The results of the current study demonstrate that the predictive value of the models developed for semidwarf varieties increased when the data of more than one crop season were aggregated to build them. For future studies, the assessment of biophysical parameters earlier during the growing season will improve the accuracy of LAI estimates, particularly when values are low, but not because of a reduction in the chlorophyll content caused by the senescence. This leads to the conclusion that more research is needed to generate series of data from multiple years and growing stages in order to improve the reliability of the predictions obtained with the models developed from the UAV 4-band multispectral (Parrot Sequoia) and RGB cameras. In addition, the use of machine learning techniques should be addressed.

**Supplementary Materials:** The following are available online at https://www.mdpi.com/2072-4292/13/6/1187/s1, Table S1: List of accessions. Figures S1–S4: ANOVA and Scatterplots for the agronomic traits. Table S2: Spectrometer calibration.

**Author Contributions:** Conceptualization, J.M.S., C.R. and J.B.; methodology, R.R., J.M.S., D.V. and J.B.; software, R.R. and J.B.; formal analysis, R.R., J.M.S., D.V. and J.B.; investigation, R.R. and J.B.; writing—original draft preparation, R.R.; writing—review and editing, R.R., J.M.S., D.V., C.R. and J.B.; supervision, J.M.S. and J.B.; project administration, J.M.S. and C.R.; funding acquisition, J.M.S., C.R. and J.B. All authors have read and agreed to the published version of the manuscript.

**Funding:** This study was funded by projects PID2019-109089RB-C31 and RTI2018-099949-R-C21 (Ministerio de Ciencia e Innovación, Spain). R.R. is a recipient of a PhD grant from the Spanish Ministry of Economy and Competitiveness.

**Data Availability Statement:** The data presented in this study are available within the manuscript and in the Supplementary Materials.

**Acknowledgments:** The authors acknowledge the contribution of the CERCA Program (Generalitat de Catalunya).

**Conflicts of Interest:** The authors declare no conflict of interest.

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
