# Peer review of "Using Unmanned Aerial Vehicle and Ground-Based RGB Indices to Assess Agronomic Performance of Wheat Landraces and Cultivars in a Mediterranean-Type Environment"

_remotesensing, doi:10.3390/rs13061187_

Round 1
Reviewer 1 Report
General comment:
The study presents the comparison of Unmanned Aerial Vehicles (UAVs) multispectral images and ground-based RGB images for assessing agronomic performances of rainfed wheat in the Mediterranean environment. The study seems interesting; however, the study has several issues in the methods applied (see below) and requires further modification/explanation of methods and subsequent results. The results reported are not remarkable (low R2 values) to warrant the applicability of model(s) for predicting agronomic traits. Please see the comments below:
Methods: Agronomic traits are not clearly explained. Add a table (similar to Table 2 and Table 3) to report the traits and explain which trait is the main focus of the study. If the focus of the study is LAI, as highlighted in the abstract, just report the LAI only and remove information about other traits. In Table 2, add formula column to be consistent with Table 3. Rearrange the methods section (or results section) to be consistent; it’s difficult for readers to grasp the information in the present form (especially in the results section).
Results: LAI was predicted by joining data from both landraces and modern genotypes (line 240); however, the data was separated while predicted other agronomic traits (line 219). This is a major difference in methods applied to predict the agronomic traits (and this information/explanation must be in the methods section). The R2 values of models for predicting agronomic traits are less than 0.50 in most cases (Table 7, Table 8), which seem relatively low. Please explain.
Author Response
Thanks for all comments.
We have tried to include most of the suggestions made by reviewer 1.

Reviewer 2 Report
This study compares the use of vegetation indices derived from a drone based multispectral camera and a groundbased RGB camera for estimation of wheat agronomic traits in two consecutive seasons.
Overall, the Introduction section is well-formulated, presenting the appropriate background information and added-value of remote sensing for high troughput phenotyping.
A few questions may be posed regarding the methodology:
- Although LAI is a major parameter to be accurately estimated, the reference method for its estimation (using the Accupar LP- 80. Decagon Devices) seems not to be extensively explained. A main concern refers to distint values for χ (canopy structure), which should be different for distinct genotypes. Other standard automatic parameters, such as geographical coordinates, time (i.e. sun angle) should have been accounted and, certainly, the readers of this paper would be more trustful of measurements.
- Variable importance in the presence of high levels of multicollinearity between features (VIs and RGB indices) may be biased and should be consider carefully. If redudant, the selecion or indication of one specific feature is non-conclusive. It is important to discuss the correlation between features prior to feature selection under the risk of selecting differnt (although equivalent) groups of variables
- Parrot Sequoia has been shown to provide poor radiometric quality (reference below) and affected by imagery processing (radiometric calibration and orthomosaic generation). Lines 183 states that georectification was perfomed in QGIS. However, how was the mosaic generated (crucial task)?
Franzini, M.; Ronchetti, G.; Sona, G.; Casella, V. Geometric and Radiometric Consistency of Parrot Sequoia Multispectral Imagery for Precision Agriculture Applications. Appl. Sci. 2019, 9, 5314. https://doi.org/10.3390/app9245314
- Figure 2: Rather than rainfall and temperature, a water deficit figure would provide a deeper understanding about the water for each season. This can be of particular importance, provided that the soils have presented organic matter levels above 3%, and, consequently, would provide a larger water holding capacity. Nevertheless, 2018 was a particularly dry year, but that could be best presented in respect to wheat water requirements (thus, the water deficit figure).
Results:
- Table 7: High feature coefficients indicate a high level of multicollinearity between features. For instance: NGm2 = ‒2167724 + 2631346GNDVI_2 + 12018GNDVI_3. As the selected regression method is a form of linear model, an in-depth analysis of the regression coefficients and (inference) can be performed. The paper/analysis would certainly benefit from this.
Discussion and Conclusion:
Remote Sensing is based on accurate measurements of physical (radiometric) units. Both instruments/methods (if not fully controlled for) have questionable accuracy. Consequently, regardless of which index may be have a higher importance/explanatory power, the question remain whether the method was not generalizable across seasons due to poor (physical) measurements.
---
Typos and minors errors.
line 35 - "G=ground-based RGB", please check whether it should only be "ground-based RGB"?
line 107 - Please also add the dimensions (width and length) of the plot.
line 108 - The seed rate was adjusted to 250 germinable seeds **per** m2
line 116 - Equation 1 (GDD) is introduced, yet no major reference is made in the remaining of the text, nor it is used to corroborate results, arguments or conclusions.
line 120 - Original fragment of the sentence: "The yield components number of spikes per square meter (NSm2), number of grains.. were obtained from.. ". Would the following wording best convery the intended message: "**The plant yield parameters, namely** number of spikes per square meter..."
line 123 - "drying them at 70 °C for 24 h". Is such time period (i.e., 24 hours) sufficient for obtaining a constant weight? Other researchers have employed (reference below) 70 °C for **48 hours**. It would be an improvement to the Methods section if a reference could be provided to justify the chosen time-period (i.e., 24 hours).
Calderini, D.F., Abeledo, L.G. and Slafer, G.A. (2000), Physiological Maturity in Wheat Based on Kernel Water and Dry Matter. Agron. J., 92: 895-901. https://doi.org/10.2134/agronj2000.925895x
line 126 - "fraction of photosynthetically active radiation (fiPAR)". Please check whether the measurements refer to the the fraction of **intercepted** PAR or
line 132 - For consideration, ref [24] J. M. Norman and P. G. Jarvis, corresponds to a LAI estimation method developed for Sitka Spruce (Picea sitchensis (Bong.) Carr.).
line 192 - "Vis" instead of "VIs"
line 232 to 235 - Table 4 and Table 5 seem to have been combined in one single table.
line 325 - The term "nebulosity' would be better replaced by "cloud cover".
ref [23] - ZADOKS, J.C.; CHANG, T.T.; KONZAK, C.F. A decimal code for the growth stages of cereals. Weed Res. 1974, 549 14, 415–421. (All names and last names are capitalized, possibly due to the bibliographic reference manager).
Author Response
Thanks for all comments.
We have tried to include all the suggestions and comments made by reviewer 2.

Reviewer 3 Report
General comments
The manuscript covers an interesting topic and suits to the scope of the journal and issue. The manuscript is well written. A minor revision is recommended with the following comments.
Comments
The title could be more comprehensive, suggested as, ‘Using unmanned aerial vehicle and ground-based RGB indices to asses agronomic performance of wheat landraces and cultivars.’ Give full form of UAV.
L13-20 – The rationale of the study given in the abstract is too elaborate. Could be concised to one or two sentences. It suits more to the Introduction section.
L35 – Correct the spelling – ground-based
L40 – The website may be given as per the journal reference style (http://www.fao.org/faostat/). It should quoted as reference [1].
L51 – same as L40.
Introduction – this section is written nicely
L110 – Correct as per the reference style of the journal - Zadoks et al. [23].
Table 1 – The abbreviations in the table title can be omitted and the full form can be given in the table itself as the table becomes self-explanatory. Giving more abbreviations obstructs the readability of table.
L156 – Use the multiplication sign in - 1152 x 768 pixels - × from insert symbol
L170 – same as L156
L198-201 – How was the data divided in to prediction and validation purpose dataset. What was the ratio e.g. 70-30% is 2/3rd – 1/3rd. What statistical approach and package was used to divide the data.
L236 – Table5 – All the abbreviations in the table may be spelled in full e.g. SS, CCV (%). The same changes may be done throughout the manuscript.
L246 – Table 6 -& Table 7 – The prediction accuracies of most of the results obtained in the study is less than R2 0.50, which is very low (except MTVI1 0.61, which is also low). As per the model evaluation parameters used mostly to assess the predictive ability of models or regression equations, predictions > 0.75 or other criteria like ratio of performance to deviation, root means square error, are considered. I feel this part is missing in the manuscript. Though the predictions are significant but are in the low category. Authors may write appropriate results and discussion in light of this.
Line 301 - The sentences should not be started with the abbreviations. Needful corrections to be done throughout the manuscript
References section needs recent studies to be referred. Most of the references are old and very few are recent. Authors can review recent studies and incorporate wherever necessary.
Author Response
Thanks for all comments.
We have tried to include all comments and suggestions made by reviewer 3.

Reviewer 4 Report
General Comment: This paper evaluated the prediction accuracies of two platforms (digital camera and UAV multispectral camera) predicting wheat LAI, yield and yield components for the phenotyping uses. The content is potentially interesting for Remote Sensing readers, but there are several problems to be solved. For instance, the methods should be described more clearly such as data splitting method. The results should be provided with scatter plots to check the relationships between observed and predicted variables derived from the authors’ model. The visual plots help us understanding actual prediction abilities or causes or the error within the cultivar types. Discussion should concentrate not on the comparison of agronomic traits between modern cultivars and landraces but that of prediction abilities. From my understanding, only the comparison between cultivar types were not the main objectives. If the difference in agronomic traits between the cultivar types were essential to discuss the prediction abilities of remote sensing, then it should be done; however, most of discussion are not relevant.
Comment 1: L166–186: According to this method, the authors haven’t make orthomosaics using photogrammetrically software such as Metashape or Pix4D mapper. In my experience, images captured by sequoia can be automatically calibrated using a provided calibration panel and sunshine sensor if photogrammetrically software is used, which might be more reliable than manual calibration. Is there any evidence that the authors’ procedure was more precise?
Comment 2: L187–189: Please correct the position of table and figure titles.
Comment 3: L198–201: I think the usage of the words ‘prediction’ and ‘validation’ may be wrong. The dataset used for training model is called ‘training’ dataset. The dataset used for tuning model’s parameters is called ‘validation’ dataset. The dataset used for evaluating actual prediction accuracy, which haven’t been used for either training or validation, is just ‘test’ dataset. Thus, ‘prediction purposes’ should be named ‘training’ while ‘validation group’ should be named ‘test’ if there were no processes of validation. Relevant tables (e.g. Table 7 and 8) and texts should be revised as well. Furthermore, the way how to split the data into training and test dataset should be specified more clearly. Was it done with random splitter? How much the ratios and sample sizes between them?
Comment 4: L232–238: Table 4 and 5 are the same one?
Comment 5: L247–248: There are no description on data aggregation. This should be clearly mentioned in Materials and Methods.
Comment 6: L294–295: Note that it is much easier to get significant improvement in model accuracies if there were large effect of year on crop yield and agronomic traits. I doubt that those regression models might have only explained the large variations between years. I strongly suggest that all results of test dataset should be provided with scatter plots between observed and predicted values. Such figures can provide information about the potential causes of prediction errors or prediction saturation in the higher values that might be resulted from shortcomings of NDVI and so on. The readers cannot follow your discussion if only the statistics has been shown. To confirm the year effects, points in the scatter plot should be made with different colors or shapes. If there were outliers, then the author can more specifically and deeply discuss the causes of large errors from the perspectives of agronomic traits such as canopy structure.
Comment 7: Table 7: The sample sizes for training (‘prediction’) and test (‘validation’) may not be correct, especially for Landraces data. According to the Materials and Method, “The collection consisted of 181 landraces and 184 modern cultivars from 24 and 19 Mediterranean countries, respectively (L102–103)”. The total number of Landraces 2017 and 2018 was 84+85=169. Why the 12 samples has not been included for the analysis? Also, it’s very tough to understand this table. To clearly show the types of data items, table borders should be separated with blank spaces between items (e.g. UAV multispectral and Ground-based RGB, Prediction and Validation). Some column names seems to be bold.
Comment 8: Table 8: Again, the sample size may not be correct. Even worse, the total number exceeded the maximum sample size. Please check it carefully. The position of title should be just above the table content.
Comment 9: L316–331: This paragraph is too redundant, and should be largely deleted or moved to Introduction.
Comment 10: L354–373: The discussion was primarily focused on the comparison between the modern cultivars and landraces. However, the aim of this paper was to evaluate the prediction ability of two remote sensing platforms. Most of sentences (L358–373) may not be necessary in this study. Please consider to delete.
Comment 11: L426–429 and L439–441: To confirm those statements, the authors should provide the scatter plots as mentioned in Comment 6.
Author Response
Thanks for all comments.
We have tried to include all comments and suggestions made by reviewer 4.

Round 2
Reviewer 1 Report
Authors reported the differences in size and structure of canopy (line 442, revised manuscript) of landraces and modern cultivars. Therefore, the estimation of LAI for landraces and modern cultivars must be separated by different sets of VIs (NDVI, GNDVI, MTVI2, …). MTVI2 is reported as the most significant predictor, was the same when separated for landraces and modern cultivars? Separate the LAI estimation and report the significant predictors/equations separately.
Data (UAV and ground-based) captured at different growth stages from two growing seasons, as reported in this study, is usually a good dataset to derive models for predicting agronomic traits. Maybe the low R2 is not due to limited data or variability but rather due to the models developed (linear); explore the curvilinear/quadratic models and discuss. Considering the number of VIs included in this study, a better correlation (than currently reported) is expected with combined data from two growing seasons (Table 8).
Looks like Table 4 was missing in the original manuscript but included the revised version. As suggested before, it’s better to present the interest variables (LAI and agronomic traits) in the methods section (not in the results section - Table 4) so it follows along with the results section.
Author Response
We have replied to all comments in the attached file.

Reviewer 2 Report
The manuscript presents a significant improvement since the initial version. Particularly, I would like to applaud the use of supplementary materials.
However, a number of improvements are required both in the description of the Methods, Results and Discussion. Furthermore, the Abstract fails to explicitly inform that most relationships between explanatory and explained variables are low. Although this study does not aim to specifically demonstrate causal relationships, most model prediction performances are (as stated) low.
As major remarks, please find below a (non-exhaustive) list of improvements that must be made prior to acceptance:
Supplementary Material - Please improve the quality of the figures. inform the unit of each axis within the figures. Although not required, usually, the observed (explained) variable is expressed in the y-axis and the and estimated (explanatory) variable on x-axis.
Also, the inputs for each graph (VIs or RGB parameters) are only presented in the text, making it difficult to the reader to evaluate and learn which variables are important/useful for each retrieval task.
Furthermore, why the relationships between predicted vs. observed should be presented for LAI (as a supplementary material as well). Particularly, as the main message on the abstract makes reference to the use of MTVI for LAI estimation.
line 118 - "collected at *ripening*" It is important to be explicit in which ripening phase (milk, soft dough, hard dough, or mature) the plants were harvested. Most likely all plants were harvested at maturity (GS above Z90), but 184 different cultivar should mature in different dates, leading to a fractioned harvest. If that was the case, it should be important to state, as a harvest in either milk or soft dough and subsequent storage for 5 months may have impacts on the validity of the method.
line 120/21 - as a suggestion, to avoid any misunderstanding of your readers, please state that biomass was measured as dry matter weight.
line 332/34. "4. Discussion. The current study demonstrates the feasibility of using UAV multispectral and ground-based RGB images for bread wheat phenotyping in a Mediterranean-type environment as well as to predict grain yield before harvesting."
Results. Provided that there were (in-situ) radiometric target available within the mosaic, it is important that the authors provide evidence/results that the radiometric correction was performed adequately. In practice, this can be done through the r-squared and RMSE of the spectral data for each band/date collection in the form of a table or through supplementary material. Such can better inform if the poor performances of models were due to method (vegetation indices vs. observed traits) or through instrumental error.
The results presented in this study show low-correlation between measurements and explained variables. Consequently, the main/initial message of the Discussion cannot be stated as "demonstrates the feasibility". On the contrary, the results have shown that there is a poor correlation, with a significant performance across seasons.
Although this is discussed in the rest of that section, it is important to explicitly state these findings at the start of the discussion.
This same remark is valid for the Abstract, in which is not made explicitly clear that the overall method yielded poor prediction performances.
---
Good luck and I sincerely trust that these are useful comments.
Author Response

(The authors gave the same response as above.)

Reviewer 4 Report
I don't have any objections for the publication.
Author Response
Thanks